# The rising trend of Metaverse in education: challenges, opportunities, and ethical considerations



Sanaa Kaddoura[1] and Fatima Al Husseiny[2]

[1] Department of Computing and Applied Technology, Zayed University, Zayed City, Abu Dhabi, United Arab Emirates
[2] Lebanese International University, Beirut, Lebanon

## ABSTRACT

Metaverse is invading the educational sector and will change human-computer interaction techniques. Prominent technology executives are developing novel ways to turn the Metaverse into a learning environment, considering the rapid growth of technology. Since the COVID-19 outbreak, people have grown accustomed to teleworking, telemedicine, and numerous other forms of distance interaction. Recently, the Metaverse has been the focus of many educators. With Facebook's statement that it was rebranding and promoting itself as Meta, this field saw a surge in interest in the areas of computer science and education. There is a literature gap in studying the Metaverse's role in education. This article is a systematic review following the PRISMA framework that reviews the role of the Metaverse in education to shrink the literature gap. It presents various educational uses to aid future research in this field. Additionally, it demonstrates how enabling technologies like extended reality (XR) and the internet of everything (IoE) will significantly impact educational services in the Metaverses of the future of teaching and learning. The article also outlines key challenges, ethical issues, and potential threats to using the Metaverse for education to offer a road map for future research that will investigate how the Metaverse will improve learning and teaching experiences.

## INTRODUCTION

Maslow was a pioneer of modern and contemporary perspectives on domain-specific requirements, although he paid less attention to reproductive urges, which were among the fundamental physiological needs. Modern advancements, however, have recommended various changes to his pyramid (*Rasskazova, Ivanova & Sheldon, 2016*). When we come to the digital age, Wi-Fi has replaced one of the lowest sub-stratal demands in the modern age, significantly impacting all five hierarchy divisions (*Logan & Everall, 2019*).

Wi-Fi can help "self-actualize" and reframe digital technology's improvements so they can tap into their incredible potential. Even after most of one's basic wants are met, one could still feel new resentment and impatience until the person acts in a way that is appropriate for him (*Stosny, 2016*). The digital technology wave that swept all age groups

Corresponding author
Sanaa Kaddoura,
sanaa.kaddoura@zu.ac.ae

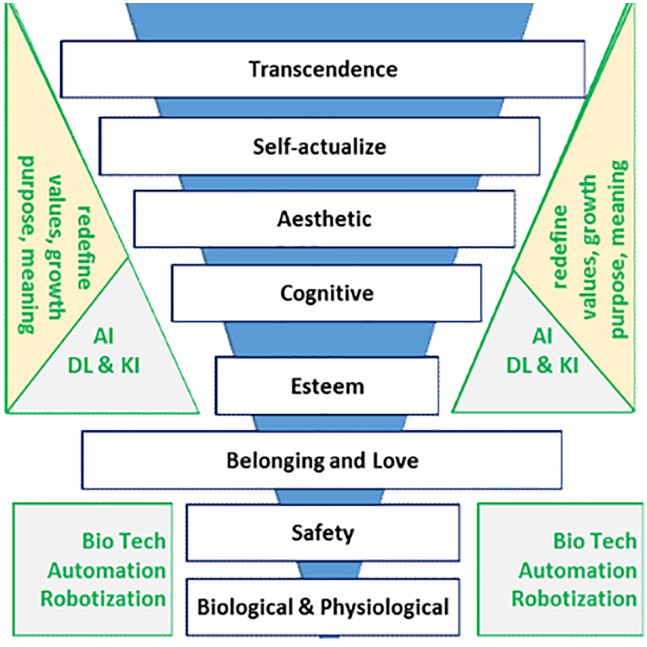

**Figure 1 A new technology-supported Maslow's pyramid of needs.**

made individuals dependent on internet usage including gadgets such as cell phones, iPads, laptops, and iPods. This led to the inclusion of the internet as one of the physiological requirements of the hierarchy of needs.

Through social networking sites like Facebook, WhatsApp, and Twitter. These demands in the digital age can now be referred to as "Love, Like, Share, and Retweet Needs" (*Bilski, 2018*).

The original pyramid's inversion is one of the key differences between the new generation's adaptations (*Medeni et al., 2020*) and interpretations. The world was completely different when psychologist Abraham Maslow's "hierarchy of needs" concept was initially published in 1943. In 2019, the world stepped on the verge of the Fourth Industrial Revolution, another significant paradigm change. This revolution resulted in a new paradigm shift for emerging digital needs.

According to recent research based on a global poll of over 43,000 people from 24 countries, which examined what it takes for an individual to succeed in today's tech-driven world, a new framework emerged (*World Economic Forum, 2019*). The digital framework of today's needs consisted of four inspired by Maslow's hierarchy of needs (1) basic needs (digital infrastructure), (2) psychological needs (social networks), (3) self-fulfillment (career and skills development), and (4) societal needs (connectedness and the digital economy) (*World Economic Forum, 2019*). Figure 1 illustrates this new dimension where AI, DL, and KI stand for artificial intelligence, deep learning, and knowledge integration, respectively (*Medeni et al., 2020*).

According to the pyramid's design (Fig. 1), the lower levels of Maslow's hierarchy give more attention to more material needs, while the upper levels allow less room for more

moral demands. The interpretation is made possible by inverting the pyramid to allocate more room to needs at higher levels and less space to needs at lower levels. This perspective offers a more precise understanding of the relationships between various human needs and is more per the socioeconomic requirements inherent in human nature. Making fresh interpretations and modifications to Maslow's models can also be beneficial. One of these new, noteworthy interpretations is part of the Digital Society Index (DSI) investigations. As a result, the hierarchy structure is maintained to offer a more unified strategy for tackling digital needs and associated societal concerns (*Digital Society Index, 2019*). For this, the new version of "Maslow's pyramid highlights the tech" generation's digital need while in the presence of virtual reality or Metaverse. Since so much of our identity is shaped by the socio-digital world, it is impossible to completely break away from it (*Serpil & Karaca, 2023*).

## RESEARCH GAP

Compared to earlier generations of the Metaverse, access to it is now possible any time and any place due to the rapid advancement of mobile technology and deep learning. This has also enhanced the accuracy of visual and language recognition, leading to more immersive environments (*Park & Kim, 2022a*). In addition, several literature reviews on the Metaverse in general (*e.g.*, *Narin, 2021*), reviews of graphics, interactions, and visualization studies related to the Metaverse (*Zhao et al., 2022*), and reflections on virtual commerce from application design and consumer behavior in the Metaverse (*Shen et al., 2021*) with scarce literature available related to Metaverse in education. This article specifically addresses the following research questions:

Q1: In what ways can Metaverse be defined?

Q2: How is the Metaverse applied in education?

Q3: What are the advantages and disadvantages of using Metaverse in education?

Q4: What are the challenges and opportunities arising from Metaverse implementation in education?

Q5: How do ethical considerations appear when Metaverse is applied in education?

## BACKGROUND

It is evident that digital reality technologies have the potential to revolutionize the domains of education, remote work, marketing, and economics, as well as the entertainment business. These technologies have also begun to establish a new paradigm for information exchange. It may be claimed that a new paradigm has developed that is centered on the idea of the Metaverse (*Mystakidis, 2022*). The terms "meta," which means beyond or after, and "universe" are combined to make the phrase "Metaverse" (*Cheng et al., 2022*). Metaverse platforms offer additional advancements across the board as virtual reality systems become more user-friendly and networked. It may be much more practical to broaden their use cases and adapt them to educational situations once virtual reality glasses and accessories obtain a more comfortable design suitable for long-term use. Making

hardware and software based on the Metaverse that may be used in educational settings is very important. Thus, collaboration and mutual guidance on this issue are crucial for educators, researchers, designers, and developers (*Sarıtaş & Topraklıkoğlu, 2022*).

Virtual reality (VR) and augmented reality (AR), two spatial, immersive technologies, are currently driving the fourth wave of computing innovation (*Kamenov, 2017*). This wave is anticipated to create the following ubiquitous computing paradigm, which has the power to revolutionize (online) business, entertainment, remote work, and education. The Metaverse is this new paradigm. Meta (a Greek prefix that means after or beyond) and universe make up the closed compound word that is the Metaverse. The Metaverse is, in other words, a post-reality cosmos that combines physical reality and digital virtuality in a continuous and enduring multiuser context. Metaverse has the potential to address the fundamental drawbacks of web-based 2D e-learning tools in the context of online distance education (*Mystakidis, 2022*).

Neal Stephenson's science fiction book Snow Crash (*Stephenson, 2003*) described the idea of the Metaverse that he coined in 1992. Stephenson is credited for introducing the concept of the Metaverse to the world (*Duan et al., 2021*). Many efforts and studies have been done to develop Metaverse technology, which is based on a fictional novel but does not actually exist as technology that can be used in real life (*Kye et al., 2021*). However, Metaverse quickly gained fame after Mark Zuckerberg made the Metaverse project official in October 2021 (*Tlili et al., 2022*).

The phrase "Extended Reality" or "Cross Reality" (XR) refers to a group of immersive technologies that create electronic, digital settings where data are represented and shown. Virtual Reality (VR), Augmented Reality (AR), and Mixed Reality (MR) are all parts of XR In all of the XR aspects, people view and engage with a totally or partially artificial digital environment created by technology. VR is a different, entirely fake environment that was produced digitally. Users experience immersion, feel as though they are in a separate universe, and behave in a manner that is similar to how they would in the real world (*Slater & Sanchez-Vives, 2016*).

This experience is enhanced by the modalities of vision, sound, touch, movement, and natural contact with virtual things with the aid of specialist multimodal equipment, including immersion helmets, VR headsets, and omnidirectional treadmills (*Pellas, Mystakidis & Kazanidis, 2021*; *Pellas, Dengel & Christopoulos, 2020*). To improve physical environments, AR takes a different method. It incorporates digital inputs and virtual aspects into the real world (*Ibáñez & Delgado-Kloos, 2018*). It geographically combines the real and virtual worlds (*Klopfer, 2008*). The result is a layer of digital artifacts spatially projected and mediated by objects like smartphones, tablets, glasses, contact lenses, or other transparent surfaces (*Mystakidis, Christopoulos & Pellas, 2021*).

The definition of MR has changed throughout time, reflecting changing technology advancements and the predominant verbal meanings and narratives. MR is a more complex notion. In that the physical world interacts in real-time with the projected digital data, MR is occasionally described as an advanced version of AR. As a result, MR remains a blend of AR and VR, the two core technologies (*Speicher, Hall & Nebeling, 2019*). In real-world settings, augmented reality (AR) is an example of a partially virtual Metaverse. A

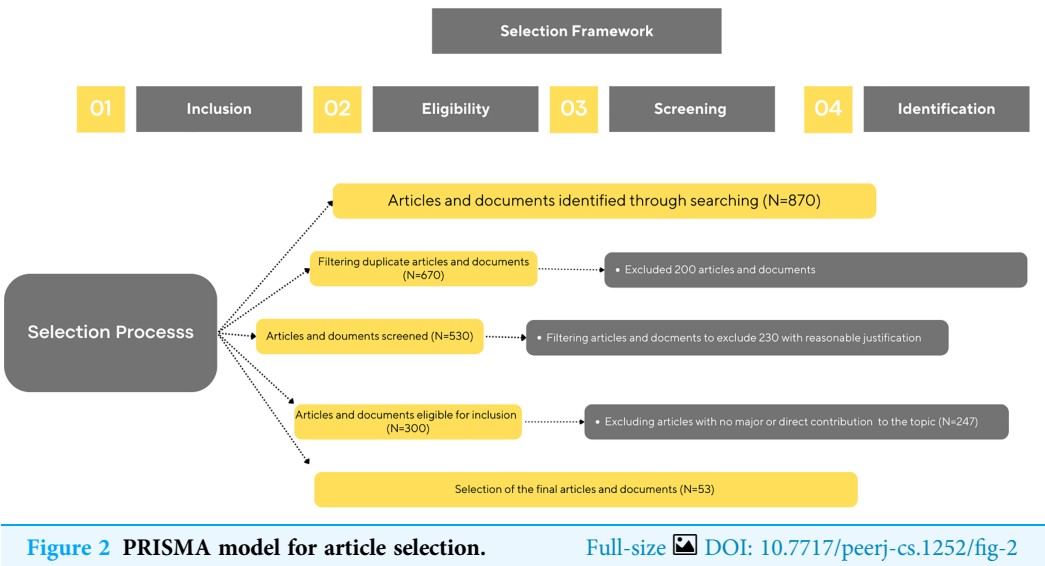

**Figure 2  PRISMA model for article selection.**

Metaverse could be entirely virtual, like a virtual reality system. People can interact socially in the Metaverse by having conversations, working on projects, playing games, and learning from their mistakes or successes (*Bourlakis, Papagiannidis & Li, 2009*; *Park & Kim, 2022a*). Real people or virtual characters could be one's companions or friends in the Metaverse (*Díaz, Saldaña & Avila, 2020*; *Kye et al., 2021*). The lifelogging feature allows for complete documentation of meta-verse life (*Thawonmas & Fukumoto, 2011*).

The Metaverse is far more than VR or AR, which some people would mistakenly believe (*Park & Kim, 2022b*). The Metaverses have a framework of three characteristics "shared," "persistent," and "decentralized,"—which distinguish it significantly from traditional VR or AR (*Min & Cai, 2022*). Additionally, artificial intelligence (AI) is a necessary technology for the Metaverse's universe to function according to the programmer's rules. From this structure, it is clear that an AR or VR system might be used to present virtual content in the Metaverse; alternatively, the Metaverse itself could include AR or VR aspects in addition to other necessary elements. People can use different identities to connect with others in a multiuser VR system such as Second Life (*Min & Cai, 2022*). Still, if the system cannot offer a persistent reality where users can live real life experiences including working, owning, learning, interacting, producing, and having fun, it is not considered a Metaverse (*Min & Cai, 2022*).

## METHODOLOGY

According to the Preferred Reporting Items for Systematic Reviews and Meta-Analyses steps (PRISMA) (*Moher et al., 2009*) which are shown (Fig. 2) and discussed below, this systematic literature review was conducted.

The PRISMA model followed to develop this review started with four steps that constructed the framework, (1) inclusion (keyword search), (2) eligibility, (3) screening, and (4) identification (*Oláh et al., 2020*).
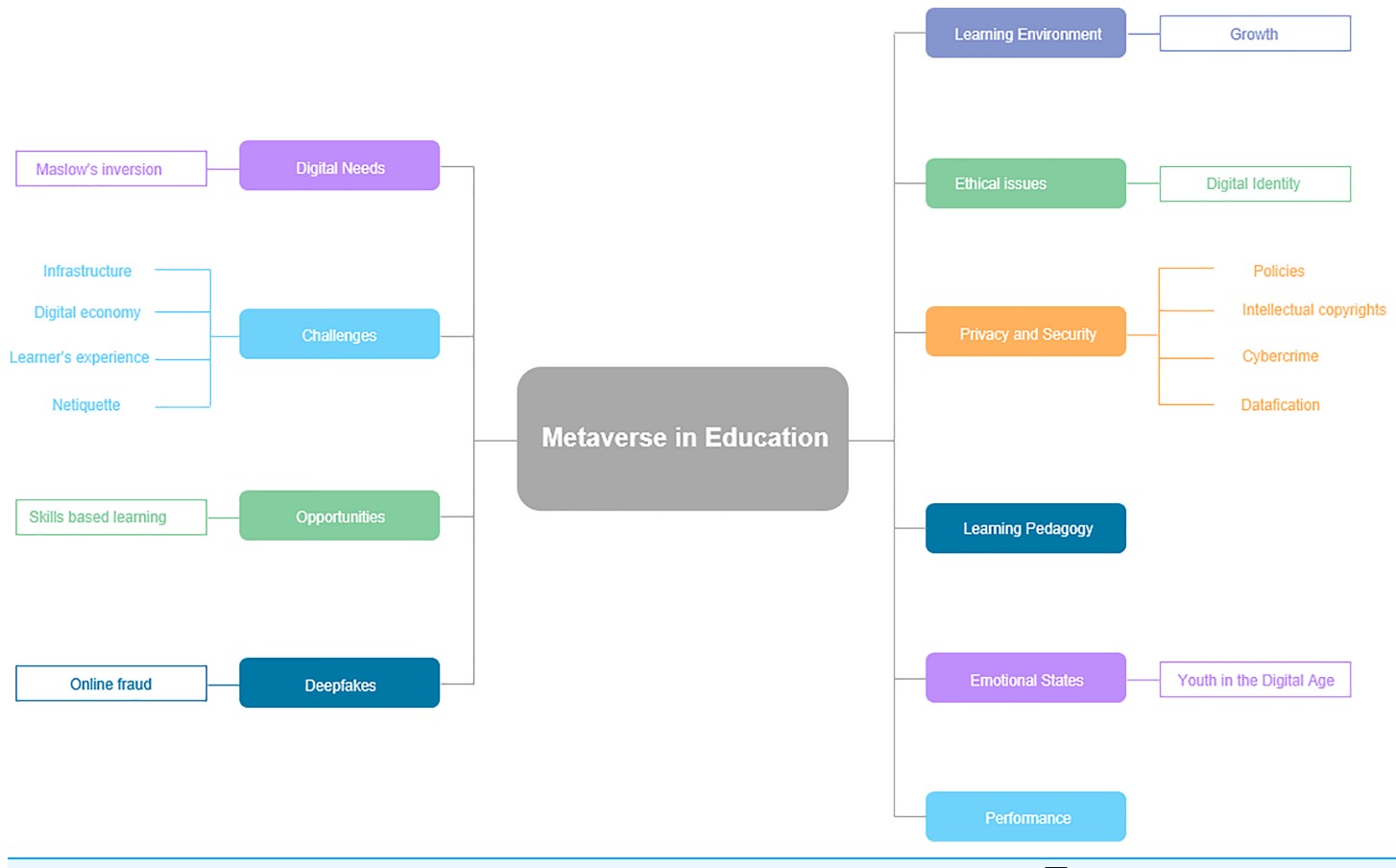

**Figure 3 Keywords search word cloud for article search.**       

## Inclusion

After a thorough search of several preview publications related to Metaverse in education, the gap in the literature was evident due to the freshness of the topic. There were a minimal number of recent articles on the issue. As a result, the research started with identifying empirical and narrowing down the search history of documents and previous work. Several database searches and search engines were used, mainly Google Scholar, ScienceDirect (Elsevier, Amsterdam, Netherlands), Taylor & Francis Online, Wiley Online Library, IEEEXplore, and Springer.

Several keywords were used to narrow down the search (Fig. 3). The keywords used were 'Metaverse and education', 'Metaverse in higher education', 'Metaverse role in education', 'Metaverse tools in education', and 'Metaverse challenges and educational opportunities'. Also, the search terms 'digital education', 'virtual reality,' and 'Metaverse' were mainly used to search the database. It is noteworthy that general keywords and phrases were excluded, for example, 'assessment, online, and remote learning.' This initial search included 870 articles.

| Table 1 Inclusion and exclusion criteria. | |
| --- | --- |
| **Inclusion criteria** | **Exclusion criteria** |
| • The article has been published in journals, books, and proceedings<br>• The report is focused on the Metaverse in education, the use of the Metaverse in education, tools, applications, challenges, opportunities | • Thesis or university papers, webinars, reviews, duplicate papers, and workshop<br>• Irrelevant literature reviews or short ones, general technical papers, articles that do not relate Metaverse to education |

### Eligibility

The research studies evaluated for inclusion in the review met several requirements listed in Table 1. After filtering the articles and documents from the initial search, 200 duplicate articles were removed.

### Screening

The screening stage was divided into two sub-stages filtering and another layer of exclusion. In the filtering stage, 530 articles were screened and narrowed down, preparing for the final exclusion stage, which eliminated 247 with reasonable justifications. In this sense, a rational explanation falls under the exclusion criteria mentioned in Table 1.

### Selection

Finally, the 53 articles selected for this review were collected and prepared for synthesis and are listed in Table 2 with their description. The final step of the selection process prepares for the discussion part.

## DATA ANALYSIS AND DISCUSSION

This section provides answers to the five questions that were raised in the Research Gap section above. A summary of the literature is presented in Table 2.

## DISCUSSION

### Metaverse in higher education

The coronavirus pandemic disrupted education globally. Both learners and teachers have had to adjust to a new, distant learning method from the beginning of 2020. Throughout the global lockdowns, technology has been essential for ensuring that education has not stopped (*EU Business School, 2022*).

The world has been severely challenged and disrupted by the COVID-19 epidemic. This pandemic has affected all facets of society, particularly the educational system, changing pupils' settings, attitudes, and abilities (*Toquero, 2020*).The end of in-person instruction in schools and colleges required legislation on a global scale, forcing the transition to online delivery methods (*Itani et al., 2022*).

Now, educators and IT professionals are thinking the same thing: if the internet helps individuals learn in its current form, what kinds of things would the Metaverse allow everyone to do? (*Contreras et al., 2022*).

The Metaverse allows the curriculum to customize to the student's needs (*Mistretta, 2022*). With new technology and schools of thought always emerging, the world is

**Table 2 Some of the selected articles and studies with descriptions.**

| Title | Year | Publisher/Journal | Key findings |
|---|---|---|---|
| Evaluation for students' learning manner using eye blinking system in Metaverse. | 2015 | *Procedia Computer Science* | New avatar-blinking technology was introduced for a virtual problem-based learning (PBL) class in the Metaverse. With the blink system, teachers may thoroughly examine students' reactions to get better results. |
| Towards data justice? The ambiguity of anti-surveillance resistance in political activism. | 2016 | *Big Data and Society* | – There is uncertainty around this type of anti-surveillance resistance in connection to more significant activist behaviors<br>– (Re)conceptualized on terms that may address the consequences of this data-driven form of governance in connection<br>– Towards more comprehensive social justice agendas<br>– Transition to surveillance capitalism |
| The geography of Pokémon go Beneficial and problematic effects on places and movement. | 2017 | *In Proceedings of the 2017 CHI Conference on Human Factors in Computing Systems, New York* | – The design of Pokémon G.O. reinforces pre-existing geographically linked biases—Pokémon G.O. has geographically-related safety risks<br>– Geographic design considerations |
| Cyber-syndrome and its formation, classification, recovery, and prevention | 2018 | *IEEE Access* | – Nomophobia<br>– Genesis stages, categorization, treatment, and prevention of cyber-syndrome<br>– How cyber-syndrome influences physical, social, and cognitive environments |
| Distributed Metaverse: creating decentralized blockchain-based model for peer-to-peer sharing of virtual spaces for mixed reality applications. | 2018 | *In Proceedings of Augmented Human International Conference (A.H.)* | – Mixed reality: social and collaborative applications<br>– Decentralized peer-to-peer distribution paradigm |
| Copyrights in higher education: motivating a research agenda. | 2019 | *Journal of Technology Transfer* | – IP ownership, copyrights, intellectual property in universities<br>– Fundamental legal and policy investigations |
| The ethics of smart city (EoSC): moral implications of hyper-connectivity, algorithmization, and the datafication of urban digital society | 2020 | *Ethics and Information Technology* | – Potential strategies for addressing the underlying moral issues by demonstrating the effects and implications of digital connectedness, algorithmization, and the datafication. |
| Identity reshaping and manipulation in cyberspace | 2020 | *Anthropological Research And Studies* | – Personal privacy<br>– Privacy of behavior and decisions<br>– Privacy of communication in an online medium<br>– AI driven decisions |
| The virtual world is a resource for hybrid education. | 2020 | *iJET* | – Metaverse as a complementary digital tool to the teaching-learning process in the university<br>– Freedom of access to synchronous and asynchronous information<br>– The issue of the design, development, and implementation of a virtual or Metaverse world in an educational environment<br>– How mobile and hybrid learning models for collaborative class. |
| Public policy for the Metaverse: key takeaways from the 2021 AR/VR policy conference | 2021 | 2021 AR/VR Policy Conference. | – Building an immersive future<br>– Innovation through accessibility and inclusion<br>– Utilizing a solid knowledge base as individuals, communities, and policy makers |

| Title | Year | Publisher/Journal | Key findings |
|---|---|---|---|
| Netiquette: ethic, education, and behavior on the internet | 2021 | *International Journal of Environmental Research and Public Health,* | – Cultural changes due to the information and communication technologies (ICT)<br>– Netiquette in the age of social network |
| Metaverse for social good: a university campus prototype. | 2021 | *In Proceedings of the 29th ACM International Conference on Multimedia.* | – Technology for social benefit<br>– Macro view, a three-layer Metaverse design: infrastructure, interaction, and ecology.<br>– Transcending toward a historical and a fictional Metaverse<br>– A blockchain-driven Metaverse prototype of a university campus |
| A complete survey on technological singularity, virtual ecosystem, and research agenda. | 2021 | Cornell University | – Accelerated digital transformation through "Metaverse"<br>– Metaverse ecosystem |
| Hidden town in 3d: teaching and reinterpreting slavery virtually at a living history museum | 2021 | *Journal on Computing and Cultural Heritage* | – Collaborative digital humanities education, including<br>– Multidisciplinary project-based learning, |
| Immersive virtual reality in K-12 and higher education | 2021 | *Virtual Real* | – Immersive virtual reality (VR) applications in higher education<br>– Virtual instructional design approaches |
| Big data, antitrust, and monopolistic power over human behavior | 2021 | *UCDavisLaw Review* | – Theory of antitrust<br>– Monopolistic hold on biopower to attain biosupremacy and control over human behavior |
| Ready teacher one: virtual and augmented reality online professional development for K-12 schoolteachers | 2021 | Computers | – Virtual reality (VR) and augmented reality (AR) stimulate students' episodic memories<br>– Educational advantages of AR and VR<br>– inquiry-based learning and digital storytelling |
| A content analysis of the Metaverse articles. | 2021 | *Journal of Metaverse* | – Gaming businesses<br>– Creating consumer's distinctive experiences and learning journey |
| The future of online trust (and why Deepfake is advancing it). | 2021 | AI Ethics | – Trustworthy AI<br>–Deepfake risk posed by unreliable AI<br>–Democracy and internet trust.<br>– Social trust challenges of the digital era |
| A survey on Metaverse: the state-of-the-art, technologies, applications, and challenges. | 2021 | Cornell University | – Technological architecture of the Metaverse<br>– Network infrastructure<br>– Management technology |
| Will Metaverse be NextG internet? Vision, hype, and reality | 2022 | Cornell University | – Essential prerequisites for the Metaverse.<br>– Possibilities and potential future areas for more innovation. |
| Definition, roles, and potential research issues of the Metaverse in education: an artificial intelligence perspective. | 2022 | *Computers and Education: Artificial Intelligence, Elsevier* | – Characteristics of the Metaverse and potential uses<br>– Metaverse definition<br>– Challenges in academic environments<br>– Functions of AI in the Metaverse and Metaverse-based schooling |
| Advancing education through extended reality and internet of everything enabled Metaverses: applications, challenges, and open issues | 2022 | Cornell University | – Metaverse paradigm<br>– Meta technology educational applications<br>– Ethical issues of using the Metaverse in education |

(Continued)

| Table 2 (continued) | | | |
|---|---|---|---|
| Title | Year | Publisher/Journal | Key findings |
| An exploratory study on the production of Metaverse ethics education contents for adolescents | 2022 | *EasyChair.* | – Virtual reality evolution during COVID-19<br>– Social and ethical issues<br>– Cybercrimes<br>– "Metaverse Ethical Principles" |
| A Metaverse: taxonomy, components, applications, and open challenges | 2022 | *IEEE Access* | – The social value of generation Z<br>– Connectedness with reality<br>– Deep learning in the Metaverse |
| The Metaverse as a virtual form of data-driven smart urbanism: on post-pandemic governance through the prism of the logic of surveillance capitalism. | 2022 | *Smart Cities* | Challenges of the Metaverse have *via* the social and economic logic of surveillance capitalism<br>– Evaluating the effects of Metaverse's widespread adoption as well as users in making decisions about its use in daily activities |
| Avatars in the Metaverse: potential legal issues and remedies | 2022 | International Cybersecurity Law Review | This article discusses problems with a real-life person's avatar in the Metaverse.<br>– What does the anticipated rise of the Metaverse entail<br>– Avatar's copyrights<br>– Methods for giving legal personality to avatars in the Metaverse \<br>– Imposing accountability on a real-life person<br>– Possible legal problems in the Metaverse<br>– Potential for statutory remedies and judicial interpretation to address avatar-related wrongdoing. |

constantly evolving. Curricula can easily fall behind, but the Metaverse has the power to turn the entire world into a virtual global school. Following the previous benefit, "gamifying" learning is one of the Metaverse's most promising potential uses in education (*EU Business School, 2022*). With its emphasis on teamwork and finishing tasks, the virtual world might make education seem like a video game, with classes organized like quests, inspiring students to complete their work and perform better (*EU Business School, 2022*).

Through immersive learning opportunities, the technology and tools of the Metaverse have significantly improved pedagogical and technical support for education, positively impacting student motivation (*Tlili et al., 2022*). In Fig. 4, technology and tools are broken into seven categories: wearable, immersive, instructional, modeling and simulation, mobile, sensors, and artificial intelligence (AI). The immersive direct experience that students receive in the Metaverse encourages teamwork and skill development in addition to engaging students in many ways in the classroom (*Tarouco et al., 2013*, *Erturk & Reynolds, 2020*). They combine virtual technologies such as Virtual Reality (VR), Multi-User Virtual Environment (MUVE), Mixed Reality (MR), and Augmented Reality required to achieve immersion (AR). Technologies that act as gateways and allow us to immerse ourselves in Metaverse environments suggest the necessity of multimodal immersion (*Mystakidis, Fragkaki & Filippousis 2021*).

*Siyaev & Jo (2021b)* demonstrates that MR is a tool that combines the physical and virtual worlds and is capable of increasing learning through deep learning voice interaction

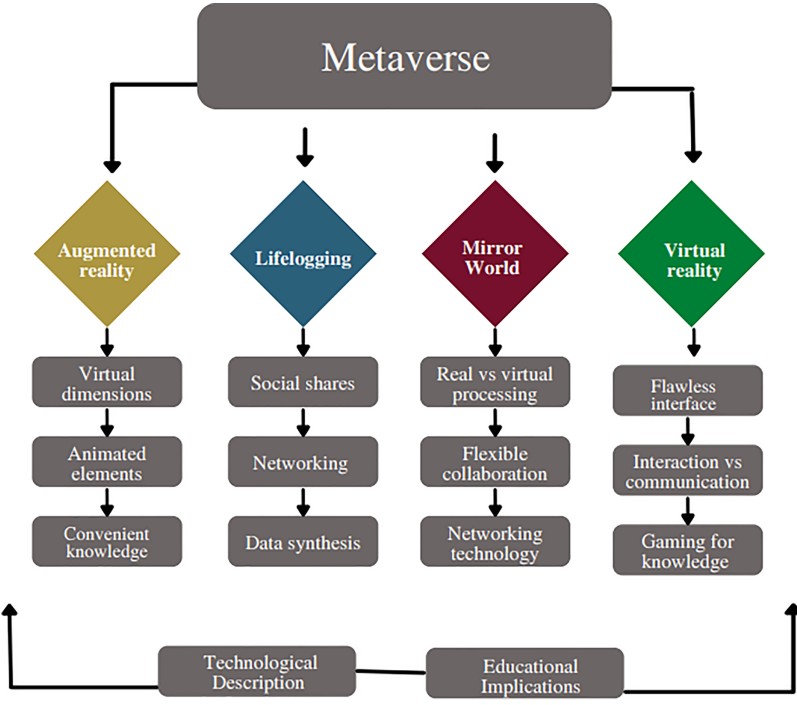

**Figure 4 The four elements of Metaverse and their educational implications.**

modules. It also mentions the usage of MR in the Metaverse. Real-time VR and AR technology enable the creation of fictitious interactive 3D characters (*Sourin, 2017*). When the Metaverse is used in education, it is combined with educational platforms, allowing the immersive settings to be a significant part of instructional topics and making it simpler to connect knowledge (*Wagner et al., 2013*).

The Metaverse in education combines the institutional learning management system (LMS), HotPotatoes, MOOC, Moodle, Teleduc, Eduquito, and Sloodle. For instance, with Web 2.0 and AVAS, Massive Open Online Courses (MOOCs) give students access to a social network. Many students may access subject information for free thanks to the Metaverse and MOOC, and online courses enable them to increase their knowledge (*Wagner et al., 2013*). Engineering, mathematics, and natural science courses employ virtual learning laboratories. The Virtual Learning Laboratory (VLL) gives students access to a collaborative, interactive, and dynamic studying environment, enhancing their motivation for learning and their education quality (*Tarouco et al., 2013*). For instance, the Moodle-created virtual environment enables the university's online learning platform to share user data, link user profiles between the two platforms, and make presentations (*Lucas, Benito & Gonzalo, 2013*).

Mobile technologies are considered the most prevalent in the Metaverse because they enable a connection between the medium and the student, through using mobile devices and geographic mobility. Learning processes can be improved when students use their avatars in Metaverse on mobile devices. One of the technology categories that enables

teachers to monitor student dynamics by evaluating student behavior is sensors and wearables, such as Microsoft HoloLens2 smart glasses and eye blinking (*Barry et al., 2015*).

Students can engage with content and issue commands in the virtual world when they use HoloLens 2 smart glasses (*Siyaev & Jo, 2021b*). The Blinking system, which primarily uses specialized software to record students' blinking times, is another widely used tool in the Metaverse. The blinking system records more blinks when a student is emotionally unstable, allowing teachers to assess the student's behavior more (*Barry et al., 2015*). In artificial intelligence, using neuro-symbolic AI, convolutional neural networks, machine learning, and semantic database technology aids students in processing learning-related material more effectively (*Duan et al., 2021*).

The core idea of the Metaverse is the analysis of complex data for understanding, regulating, and planning, and the development of artificial intelligence can be used as a foundation for processing this data (*Duan et al., 2021*). Neuro-symbolic AI can use automatic voice recognition metrics to provide feedback on user data by combining neural networks with traditional symbolic reasoning (*Siyaev & Jo, 2021a*). In training and instruction for aviation maintenance, neuro-symbolic AI is frequently employed. For instance, in courses on aviation maintenance, neuro-symbolic AI can take the place of subject-matter specialists by offering technical advice and all the resources to support efficient training and instruction in aircraft maintenance (*Siyaev & Jo, 2021a*).

Convolutional neural networks (CNN) frequently process auditory information and the command and language recognition learning and classification components, increasing learning efficiency (*Siyaev & Jo, 2021a*). The potential of the Metaverse's and Artificial Intelligence also open new roles for intelligent non-player characters (NPCs), including tutors, peers, and tutees (*Hwang & Chien, 2022*). Future research has the potential to make use of Artificial Intelligence. technology to examine how students behave and interact, as well as how well they function in the Metaverse, to create new positions. A variety of technological fields have been utilized in Metaverse to build a healthy learning ecosystem. However, several cutting-edge technologies are still not being used. For instance, a learning system with built-in anti-cheating might be developed using blockchain to increase user security. Whereas cryptocurrency is often employed in the Metaverse as a whole, this is not the case in the educational Metaverse. Using various sensors and devices, Internet of Things (IoT) technology could also be used to develop a more immersive learning environment that combines the real and virtual worlds.

The Metaverse offers a fully immersive educational experience to students. When studying planets, for instance, the Metaverse could show the entire galaxy with the ability to zoom in and out so that a learner can observe the texture and features of the cosmos. When studying old architecture, a learner can travel back in time and observe the architecture's intricate elements firsthand (*Ning et al., 2021*). The Metaverse can be conceptualized as a place where virtual reality is used to supplement the physical world with the four types of Metaverse (see Fig. 5): Augmented reality, Lifelogging, Mirror world, and virtual reality (*Kye et al., 2021*). The real world is either interconnected with virtual reality, or transformed into a different environment inside the virtual reality. From an evolutionary perspective, the Metaverse reflects a world that has spread and developed in

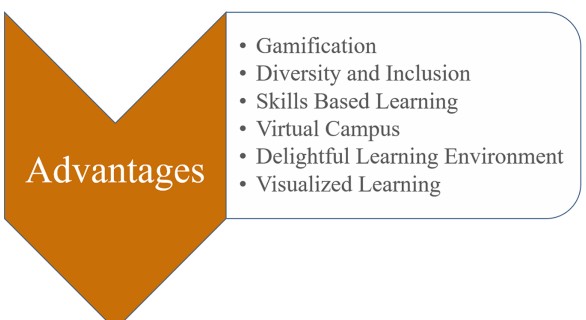

**Figure 5 Advantages of using Metaverse in education.**

reaction to the COVID-19 pandemic and is a combination of the accelerated innovations with 5G and virtual convergence technology (*Sun et al., 2022*).

The four different forms of the Metaverse first evolved independently. However, through interacting and dismantling barriers, they have later developed into a new type of convergence/composite service. For instance, the lifelogging service Ghost Pacer mixes augmented reality and lifelogging (*Lee, 2021*). Virtual reality is the most varied and actively utilized Metaverse among the four forms of education. The current non-face-to-face period has been distinguished in particular by the widespread use of virtual reality, which can be accessed from anywhere regardless of distance or geography.

The "5 Cs" are the following qualities of the virtual reality Metaverse that set it apart from already available platform services: First, as a canon, designers and participants jointly create and expand the space-time of the Metaverse. Second, anyone in the Metaverse can create content. Third, as a form of digital currency, production and consumption are made possible through the production of various ranges. Fourth, the Metaverse ensures the continuity of daily life. Fifth, the Metaverse connects the real and the virtual, connecting the Metaverse world (avatars) (*Kye et al., 2021*).

The "digital native generation" leave social and economic imprints through their many online personas and avatars. The traits and educational consequences of each of the four categories of Metaverse from an academic perspective are presented in Table 3.

## Advantages and disadvantages of Metaverse in education

3D virtual worlds, primarily utilized for gaming and pleasure before Metaverse, were also used in teaching after a while, as happens with every new technology (*Bulu, 2012*). Combining education and entertainment makes learning environments more appealing, encouraging students to participate more actively. As a result, the Metaverse is now being used as a 3D virtual environment in educational activities. However, it is progressively becoming clear that the idea of the Metaverse is not without its issues and limitations (*Minocha & Reeves, 2010*). Maintaining an active learning environment emphasizes improving students' learning abilities, particularly critical thinking and participation in the process. Although the idea of critical thinking is crucial in online learning settings, it is not adequately investigated. New frameworks are required to enhance learning abilities and online performance (*Kaddoura & Al Husseiny, 2021a*, *2021b*).

**Table 3 Educational implications of the four elements of the Metaverse.**

| Augmented reality | Lifelogging | Mirror world | Virtual reality |
|---|---|---|---|
| – Accurately solve problems using virtual digital information to teach invisible portions graphically<br>– Deep grasp of subject matter | – Reviewing and reflecting on one's daily activities<br>– Capacity to present and use the information correctly<br>– Retrieving feedback from others on social media sites results in reinforcement and incentives<br>– Use collective intelligence to recreate the information learned<br>– Examine knowledge understanding and make improvements based on learning analytics data<br>– Data-driven decision for teachers for personalized learning and guidance | – Learning beyond physical borders<br>– Conducting collaborative real-time online classes<br>– Students can achieve "learning by making, real-life experience"<br>– Engaging with their digital heritage. | – Practice can be done through virtual simulation<br>– learning beyond time<br>– Improving strategic and comprehensive problem-solving skills |

Because there is a literature gap regarding Metaverse in education, the benefits of using it in educational settings are listed in Fig. 5 and explained below.

### Gamification

Generation Z individuals are the primary users of the Metaverse. Generation Z refers to people who were born after 1995 and who differ from earlier generations in many ways. As this generation grew up during the development of PC and mobile gaming technologies, scientists focused on their preference for game-like experiences (*Park, Min & Kim, 2021*). The term "gameful experience" refers to a method that applies game mechanics like points, badges, levels, and leaderboards to non-game contexts including business, education, and healthcare in order to increase user engagement and motivation (*Deterding et al., 2021*). Through these game features, gamification in education promotes learner motivation, involvement in learning, and attitude improvement (*Kim et al., 2018*).

### Diversity and inclusion

The use of Metaverse in education enables students from different backgrounds to learn in the same environment (*Park, 2021*). Because of several socioeconomic circumstances, the culture of diversity, equity, and inclusion has become more prominent in a lot of businesses. Access to a technology or service has made it possible to establish a baseline of social equity that supports the promotion of diversity, equity, and inclusion (*Zallio & Clarkson, 2022*). The Metaverse presents an excellent opportunity to overcome many of the difficulties people encounter in the real world, including physical disabilities, restricted access to certain locations, social exclusion brought on by a lack of exposure to the diversity of people, and opportunities to provide new sensory experiences for those with sensory or cognitive impairments (*Zallio & Clarkson, 2022*).

### Skill based learning

Learning in the Metaverse offers the chance to practice various skills and repeat them as often as desired for skill-based learning approach (*Locurcio, 2022*; *Koo, 2021*). In addition, the current augmented reality mobile media approach can foster the growth of cognitive

abilities and produce innovative experiences in the acquisition of material. The main objective is to boost students' academic performance, and this will highlight how mobile augmented reality learning resources may be very useful when blended into classroom instruction (*Marini et al., 2022*).

As there are many obstacles, advances in self-education necessitate technological and educational validation. Without extensive prior training, education, and preparation of teachers and pupils, it is vital to ascertain whether such a move will increase the quality of instruction (*Pedro, Piedade & Filipe, 2019*). For this activity to improve education quality, teachers must create communication channels that are appropriate for digital needs and comprehend the role that technology plays in modern learning. To compete in upcoming international economic competitions, students must also advance their technological proficiency (*Olszewski & Crompton, 2020*). To compete in the global market, students need to develop their technological abilities. Just a few of the skills pupils need to develop include critical thinking aptitudes, inventive problem-solving abilities, communication aptitudes, cooperative aptitudes, and tech-savvy citizen aptitudes (*Marini et al., 2022*).

### Virtual campus

The World Bank Group, through its Open Learning Campus, stresses the fact on more merits of Metaverse in education, such as connecting learners through a realistic virtual campus, improving real-world expertise in hybrid and virtual situations, investigating various worlds *via* storytelling and visualization, increasing human talents in social or challenging circumstances, developing accessible solutions for learners with disabilities and increasing learning performance (*World Bank Group, 2022*). The majority of current efforts to create a campus Metaverse concentrate on virtual environments or virtual reality (*Duan et al., 2021*). An AR Metaverse allows situated interaction in a specific physical space, bridging the gap between the physical campus infrastructure and digital content. To expose located services to campus users, an AR campus Metaverse offers a rare chance to combine digital content with physical infrastructure (*Braud, Fernández & Hui, 2022*).

### Delightful learning environment

A Metaverse's social involvement is the first thing to be considered while deploying it (*Getchell et al., 2010*). The four fundamental components of an educational scenario are teachers, students, learning resources, and learning environments (*Xu et al., 2019*). In an educational Metaverse, virtual learning environments are not fixed and could be replicated and modified by some or all of the consumers who inhabit them, facilitating interaction between students and learning resources and environments (*Cai, Jiao & Song, 2022*). In fact, the Metaverse facilitates the interaction between the actual world and the virtual world for learning by doing and living through offering realistic and rich educational experiences for learners (*Locurcio, 2022*; *Park, 2021*; *Lee & Hwang, 2022*). The Metaverse can significantly help realize concrete learning which isone of the most desired roles in instructional environments. This appeals to students' visual, aural, and tactilesenses which will create a delightful learning environment (*Locurcio, 2022*; *Koo, 2021*; *Lee & Hwang, 2022*).

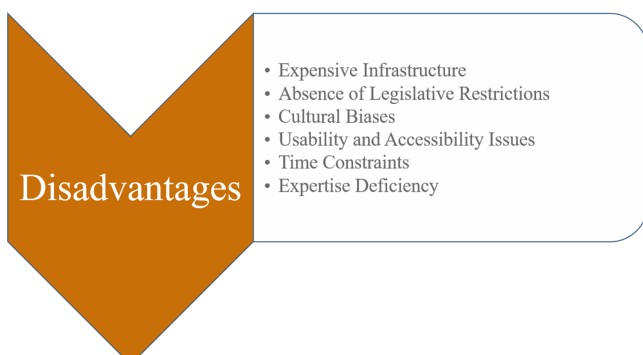

**Figure 6** **Disadvantages of using Metaverse in education.**

### Visualized learning

Due to the quick advancements in computer vision and graphics technology, the Metaverse scenes have a strong visual impact. Teachers and students should be able to view realistic virtual scenes, and their teaching actions should interact very organically, in an effective learning Metaverse educational setting. Therefore, the educational Metaverse cannot disregard the advancement of natural interaction technologies in the teaching and learning environment in addition to the complete development of visual experience technologies (*Cai, Jiao & Song, 2022*).

On the other hand, there are drawbacks to the Metaverse use in educational settings, as illustrated in Fig. 6 and explained in the section below.

### Expensive infrastructure

In addition to costing more than traditional learning tools like computers and books, XR technologies frequently call for a fast internet connection. Furthermore, creating content costs more since it requires specific tools and abilities to produce the interactive virtual environments needed for efficient learning (*Pimentel et al., 2022*).

### Absence of legislative restrictions

The Metaverse environment is now regarded as unusual because it is a relatively new notion. There are no legislative restrictions in this field at the national or international levels. Therefore, making legal arrangements for this circumstance is essential and may interest the national and international educational communities. Since the Metaverse is a decentralized Internet environment, its computer security appears lacking (*Discover Hub Pages, 2022*; *Baltaş, 2022*).

### Cultural biases

The widespread use of ICT raises concerns about the possibility of widening social and economic disparities since it may alienate "the unskilled and IT illiterate sectors of the local poorer community" (*Hollands, 2008*). In virtual reality, a number of inequities in reality will be infinitely enlarged. The Metaverse enables the reshaping of social equality laws (*Keshavarzi et al., 2020*).

### Usability and accessibility issues

For many people, the XR technologies available today are difficult to utilize. For instance, a person with restricted hand movement can find it difficult to utilize controls. Wearing glasses, for example, can make using an HMD challenging. Additionally, not all persons from different origins and identities can access certain technology equally. For instance, certain head coverings and hairstyles, such as religious head coverings and natural Black hairstyles, cannot be worn over headphones (*Pimentel et al., 2022*). XR is particularly suited to teaching skills, competences, beliefs, and attitudes that multiple-choice learning tests might not be able to capture since it can give a more personalised and unstructured learning experience. Although XR allows for the collection of a variety of user data types (such as movement and eye tracking), this data type has not yet been transformed into useful learning assessments (*Pimentel et al., 2022*).

### Time constraints

The effects of MDs used in VR and MR might deteriorate with time and can cause discomfort or nausea in users in as little as 10–15 min, hence it is better to utilize them sparingly rather than as the primary educational modality for an entire course. For longer-term interventions, different technologies like computer programs, augmented reality on mobile devices, or low-tech fixes may be more appropriate (*Pimentel et al., 2022*).

### Expertise deficiency

The learner's perception of how well technology fits into their background, standards, and previous activities is known as perceived compatibility. In other words, compatibility refers to how the students see the advantages of utilizing technology (*Greenhalgh et al., 2004*). This premise is consistent with the notion of compatibility as people's willingness to accept novel technological advancements when those advancements are compatible with their established norms and predictions (*Rogers, Singhal & Quinlan, 2014*).

## Challenges and ethical considerations

The ethical implications of the Metaverse are being discussed more than ever now, as new technologies have always been required. For the Metaverse to be a secure virtual space that transcends the current communication and equity barriers, many ethical issues need to be addressed.

## Device challenges

Processing, storage, and data transfer are the three factors that makeup computing power: the capacity to process data. In the era of the digital economy, computing power is a crucial resource, and its infrastructure is a vital support for technological advancement (*Ning et al., 2021*). Metaverse entails a more significant user base, better network resources, and more processing power (*Ning et al., 2021*). The emergence of new business models and the cloud computing-based Metaverse platform has raised the need for computer resources and opened up new opportunities for their growth. High demands are placed on client device performance and server resilience by the cloud storage, cloud computing, cloud

rendering, and other technologies employed by Metaverse. Metaverse must constantly increase processing complexity, speed, and power consumption (*Ning et al., 2021*).

The price of VR headsets and other technological innovations that will enable individuals to immerse themselves in the many virtual worlds of the Metaverse is considered a difficulty that may present barriers to widespread acceptance. Examples include Horizon Home, Future of Work, and Presence Platform. *Johnson (2022)* gives up-to-date data on how much consumers would shell out for cutting-edge VR equipment to let them feel what they are experiencing in the Metaverse. Additionally, specific segments of society will likely be locked out of the Metaverse based on factors like education, socioeconomic status, age, and disability.

The functioning of Metaverse will produce a significant amount of data, including sensor-generated metadata, a shared virtual area for user social interactions, and the transmission of high-resolution video feeds, needing significant network capacity. The Metaverse, however, might not be supported by the current 5G technology. Although theoretically, 10–20 Gbps is the most incredible speed that 5G can provide, the bandwidth needs of the Metaverse may exceed what 5G can provide. Peer-to-peer (P2P) communication methods could be a solution to this problem. User devices must therefore mix the content obtained from various parties and present the virtual environment consequently (*Cheng et al., 2022*).

The user experience is critically dependent on latency. Since users may access the Metaverse from different locations, it can be challenging to guarantee low latency in geographically dispersed areas. While this is going on, sensors in the Metaverse, like those on haptic and XR headsets, need latency as low as tens of milliseconds to maintain an immersive user experience (*Zhang, Han & Hui, 2022*). The delay between a user's motion and how others perceive their reflection in the Metaverse is a crucial measure to optimize, much to the motion-to-photon latency in virtual reality. The network edge's remote rendering (*Koller et al., 2004*) of virtual content is a potential strategy for lowering the latency above.

## Social and ethical considerations

People developed a new identity in the Metaverse, which has also produced a unique environment in which to live and interact. There are more nuanced social dynamics there. The Metaverse must govern and restrict user behavior and create explicit ethical and moral guidelines to maintain a positive ecological environment.

The moral and ethical problems that Metaverse needs to address are:

- Issues with integrity include printing and spreading false information and fraud;
- Intellectual property rights violations.

The importance of ethics at this moment has increased significantly (*Ning et al., 2021*). The original code of ethics has been impacted, and the new code of ethics is being developed at a slow pace that cannot keep up with the growth of the Metaverse. As a result, the Metaverse's oversight should be improved, and pertinent rules and regulations should be created and promptly updated (*Ning et al., 2021*).

As a result, the Metaverse will continue to reinforce the digital divides and favoritism that already exist regarding access to current social media platforms (*Rosenberg, 2022*). However, the prevalence of spam content on social media is drastically rising, making spam identification essential. As more individuals utilize social media, such as Facebook, Twitter, YouTube, and email, the amount of spam content also increases. Particularly during the pandemic, people are spending an excessive amount of time on social media. Through social media, users receive a lot of text messages, and they often fail to notice the spam content contained in these messages (*Kaddoura et al., 2022*).

Technology use is currently subject to accessibility stratification and discrimination, with tech-savvy individuals leading the way in its uptake. This will prevent all residents from having full access, incredibly the less tech-savvy, those who grew up before laptops, and those who are handicapped by universal design. Despite Meta's assertion that using new technology will help young people develop their critical thinking skills, this has always been the case regarding its application. The main concern is whether the digital divide in virtual/augmented cities could develop into a Metaverse split. However, this might benefit social groupings with unfavorable attitudes and feelings toward the Metaverse. This is supported by the significant societal issues that the extensive usage of social media platforms has brought about for many communities (*Rosenberg, 2022*).

This raises the question of whether humans will broadly embrace the Metaverse. This is a challenging subject to answer because people do not always embrace new experiences that are technologically possible. The Metaverse probably will not appeal to all social classes, at least not in the same way or at the same time. The unequal distribution of socioeconomic risks and advantages in society and individual choices and beliefs is the cause. According to *Lee et al. (2021)* the sustainability of the Metaverse will depend on the social acceptance of the Metaverse as well as several impacting design aspects, such as fairness, cyberbullying, gadget acceptability, avatar acceptability, and privacy risks. In particular, it is vital to apply a new socio-humanitarian logic in the construction of the Metaverse, as claimed by *Gurov & Konkova (2022)*.

Similarly, AmI's, Ambient Intelligence, a vision was determined by humanistic design issues, see *Bibri & Zaheer (2022)* for a detailed discussion of the seminal role of participative and humanistic design for the sustainability of AmI. AmI has emerged in the past 10 years as a multidisciplinary field within ubiquitous computing (*José, Rodrigues & Otero, 2010*). The development of AmI required a comprehensive perspective that considered people and other non-technical issues. As a result, AmI's vision is constantly being rebuilt around fresh concepts, ideas, and points to address, and R&D has changed its emphasis to creating workable win-win solutions. Furthermore, the Metaverse will likely give rise to new issues that could limit everyone's ability to access and use its goods, services, and technology or lead people to reject it if new skills and knowledge are needed, which will lead to varying degrees of acceptance (*Johnson, 2022*). Increased technological literacy and skill levels were only 34% of the benefits of the Metaverse globally in 2021, according to recent statistics (*Johnson, 2022*).

This percentage also shows that there are still a significant number of people who will not profit from the Metaverse, including some who will be unable to use it effectively

because it is a more complex platform than current social media. This is in line with findings from earlier research on the vision of AmI, which was conducted. Adults in the United States reported having the following attitudes about the Metaverse as of January 2022: curious (33%), uninterested (27%), suspicious (23%), concerned (19%), indifferent (19%), excited (18%), hopeful (16%), bewildered (12%), and none of these (7%) (*Johnson, 2022*). As a result, unpleasant emotions frequently outweigh happy ones. This could be explained by the increased awareness of social media platforms within this sector of society during the previous 20 years.

Therefore, it appears that this community group is not expressing any desire for the extraordinary in service encounters in the virtual world, which might not help promote what is envisioned claimed to be desired, pleasant, and amusing to specific groups of society in the short term. One of these groups is the young people, who appear to be interested in and enthusiastic about the Metaverse, with unpredictable medium-term results. To encourage more people to connect the potential outcomes to their actual needs, dreams, and ambitions, the Meta-verse should at the very least expose what the so-called post-reality cosmos has to offer even though the Metaverse provides the following advantages globally (*Johnson, 2022*). Avatars can participate in heterogeneous real-time activities that theoretically involve an infinite number of concurrent users across numerous virtual environments (*Lee et al., 2021*; *Bibri & Zaheer, 2022*).

## Network security and privacy concerns

The true identity and the Metaverse are tightly related to one another (*Ning et al., 2021*). Like the previous network environment, the Metaverse must fully care for data privacy protection issues as it builds a new generation of networks (*Ning et al., 2021*). As worries about security in immersive educational platforms increase, attacks on head-mounted display (HMD) devices could give attackers the ability to confuse users by superimposing or altering the pictures in their field of view (*Jagatheesaperumal et al., 2022*).

As a matter of fact, the privacy and security concerns in the Metaverse demand our attention. Although our measurement investigation confirmed that commercial social VR platforms use secure communication protocols (such as TLS and DTLS) to protect sent data, the Metaverse may still raise various security issues, including user identifying data (*Mathis et al., 2021*). Due to the need for headset access, users frequently must identify themselves using biometric data, making them vulnerable to security assaults (*Mathis et al., 2021*).

The Metaverse's digital twins also require our defense. In this sense, many intricate machine learning models will support digital twins, impacting real-world things. Attacking these theories will have unpredicted consequences for the physical universe. Additionally, user-worn headsets would continuously gather personal data (such as biometric data and user activity) to enhance the QoE, raising privacy concerns. Blockchain storage for digital twins from the Metaverse and biometric data is one conceivable direction (*Ryskeldiev et al., 2018*). However, users of the Metaverse should be most concerned about privacy risks, mainly since the privacy-enhancing mechanisms presented so far are insufficient to

address this unique problem. Technology can only protect privacy to a certain extent; even this capability has built-in constraints.

The Metaverse faces a severe difficulty and conundrum about confidentiality (*e.g.*, *Dick, 2020*, *Leenes, 2007*; *Lee et al., 2021*; *Falchuk, Loeb & Neff, 2018*). Acceptable procedures for gaining access to and sharing personally identifiable information about individuals are still up for discussion. Given the enormous amount of data that will be gathered, analyzed, classified, commoditized, and commodified, datafication and privacy are at the foundation of the Metaverse. For businesses, the value of data lies not in its physical presence but in the connections, it can make with databases and analytical tools (*Zuboff, 2016*). In the shape of new networked, digital technologies that have permeated the fabric of daily life, smart cities are undergoing an unheard-of escalation of datafication and algorithmization. This raises severe issues with the invasion of privacy. In the context of intelligent city ethics, *Calvo (2020)* discusses the moral implications of the datafication and algorithmization of urban society.

In terms of legislation and information practice standards as a matter of justice, all harms will present considerable hurdles to the Metaverse's privacy protection measures. Fairness is a crucial concern regarding the rules and principles used in the Metaverse to profile and socially classify persons in the virtual world behaviorally. The Metaverse should prioritize algorithmic fairness in all of its designs. As a result, procedural justices should continue to serve in governing capacities since they "demand a high degree of openness to the users and outcome control mechanisms" (*Lee et al., 2021*). Whatever the case, algorithmic governance involves unfairness and inequity that replicate data justice issues (*Dencik, Hintz & Cable, 2016*; *Taylor, 2017*) across various demographics (*Benjamin, 2019*; *Noble, 2018*) with potentially harmful outcomes.

Due to digital illiteracy, psychological manipulation, cognitive dissonance, and privacy paradox, the problems, and hazards of privacy invasion do not seem to be a barrier to the youth age group using social media as a daily activity (*i.e.*, users willingly share their information). It follows that the acceptance of the Metaverse by the same group is most likely to be the case. But the adult group is more concerned with privacy (*e.g.*, *Johnson, 2022*). Coming up with verifiable privacy measures is one of the main issues that the Metaverse needs to address to be accepted by society (*Lee et al., 2021*).

## Physical and mental health

The problematic use of the Metaverse and user addiction are challenging issues to address and overcome. The COVID-19 pandemic and its long-term effects on urban living and working conditions only exacerbate this. A significant body of recent research has shown that the continuous use of such forms could produce further problems in terms of the abusive use or addiction to the internet, taking into account the shift caused by this crisis from face-to-face meetings or social gatherings to virtual forms (*García, Cruz Sánchez Gómez & García-Valcárcel Muñoz Repiso, 2020*). Recent data and facts indicate mental health problems and addiction to virtual reality, which account for 47% of the dangers of the Metaverse based on Internet users globally in 2021 (*Johnson, 2022*). Complete reliance

on the Metaverse as a result of immersive experiences will lead to physical and mental health issues, which will further impair and cause users to do nothing in their daily lives.

Indeed, the excessive use of Metaverse products and services has led to an addiction to immersive technologies, which has sparked debates in the scientific, technological, and medical communities. It is anticipated that this topic will be the subject of more investigation, discussion, and criticism and be approached from many angles (*Ning et al., 2021*). This suggests that it will be challenging to establish and practice evidence-based initiatives and recommendations for policy makers. An individual can hang out with friends, work, play, learn, shop, create, and more in the Metaverse, claims Meta. It's not always about using the internet more. Making online time more purposeful is the goal. This contradicts itself since, at least for the younger members of society, how cyberspace will be constructed in terms of the variety of activities that may be undertaken there will most certainly drive constant usage of immersive technologies. To put it in another way, what the Metaverse entails shows the opposite. It will enable multiple avatars of real people to interact with one another and with aesthetically pleasing objects in various visually appealing virtual worlds, luring young people to spend more time online pointlessly. Many immersive elements, such as peak experiences, optimal attention span restoration, and happy emotions, are directly linked to the psychological states that may happen when people engage with the natural world in VR or augmented reality (*Bibri & Zaheer, 2022*).

Designers can imitate the effects of real-world environments on emotional states through common life scenarios in VR and AR The rehabilitative effects of simulated everyday situations in immersive environments are probably to rise. Users may also encounter super-realism, which enables them to engage in several activities in environments that closely mirror those found in reality. The Metaverse portrays bleak visions of the future even though it can develop into something engaging for some sectors of society (*Colley et al., 2017*; *Tong et al., 2017*). This is evident in how it conceptualizes the digital world, configures users and their social relationships, and ignores social and ethical principles (such as integrity, truthfulness, and loyalty) (*e.g.*, freedom, respect, responsibility). Unfortunately, many values essential for social health regarding community participation and a sense of belonging are not practiced in today's digital world. VR might cause significant user behavior changes, affecting society (*Colley et al., 2017*, *Tong et al., 2017*).

Despite this, the science fiction stories are developing the premise that the Metaverse will be present in the various scenarios of people's everyday lives, creating a dramatic shift in that life. In this regard, *Johnson (2022)* provides examples of activities that people will be able to engage in in the Metaverse that they would not be able to in the real world, such as:

– Alter consciousness with the aid of VR

– Create an alter ego of the opposite sex or different age

– Spend a lot of money on group clothing or accessories

– Play adult games that feature extreme violence and sex

– Carry out unethical experiments on virtual humans

– Watch virtual executions

These things shed some light on the types of extreme behaviors that the Metaverse may encourage or lead some individual users to engage in, which may have detrimental effects on societal norms. These include the capacity for positive assertiveness, being true to oneself in all circumstances, participating with others in one's community, treating others with respect, and communicating with others directly and honestly without purposefully hurting their feelings. Cyber-dystopias depict a society in which human traits like courage, honesty, kindness, self-awareness, and wholeheartedness are absent from humans, leading to dehumanizing experiences. Overall, the Metaverse is a collection of highly realistic virtual environments that will allow users to experiment with things that are impractical in real life (*Lewis & Taylor-Poleskey, 2021*).

The time has come for re-establishment when previously assumed rights and values encounter a new paradigm. It can be extremely harmful when technology develops, and variety is emphasized more and more in the world. Due to this ambivalence, the ethical framework must be flexible enough to keep up. Adolescents who are the primary Metaverse users are introduced to ethical education on moral metacognitive ability, moral sensitivity, moral judgment, and moral responsibility as the core values of education, including human dignity, protection of human rights, non-infringement, and commitment (*Kim & Park, 2022*).

## Netiquettes considerations

ASL (age, sex, locality), social position (occupation, wealth, standing, *etc.*), or physical interaction are not inherently linked to a virtual community. As a result, its members typically gather in person rather than having to travel great distances. Therefore, online communities are not an exception. Etiquette is a protocol for how to interact with other members and communicate information. Netiquette violators are flagged to other users and may endure a robust label ascription process (*Gabriel, 2020*). Particularly, digital etiquette is typically referred to as netiquette, where correct communication is essential for asking and answering questions respectfully and proactively by introducing oneself properly, posing precise questions, and concluding formally. Ethical online citizenship is necessary for the Metaverse-based education programs used in high schools, colleges, and businesses (*Jagatheesaperumal et al., 2022*). Indeed, netiquette is needed in online learning environments where information and communication technology (ICT) has altered socialization and communication practices. Regarding social interaction, the Metaverse may encourage individuals to experiment with their identity, role-playing, and self-expression. The Metaverse Roadmap states that this might result in new viewpoints on social norms related to gender, ethnicity, species, socioeconomic status, manners, and collective ideals and goals.

## Intellectual copyrights

With the proliferation of multiple online educational platforms, it is challenging to manage the intellectual property rights of educational content to prevent infringement on the

copyrights of learning resources (*Jagatheesaperumal et al., 2022*). Using both private and public blockchains, the digital copyright management system (*Rooksby & Hayter, 2019*) ensures the sharing of multimedia instructional content. It guarantees the privacy of educational content, making it a suitable option for Metaverse platforms. As the Metaverse is formed, develops, and scales, the IP issues from the previous economy will likewise be put under stress. Only a few of the potential problems include ownership, protection, piracy, and concerns with patents and IP definitions. The new economy will necessitate new ways of thinking in a world where physical boundaries dissolve and territorial rights assume new significance. IP produced by AI and Avatar may be the subject of legal challenges, for instance, contesting its legality.

Owners of content must be aware of their licensing restrictions and usage rights. Due to the use of famous impersonators in video games, there has already been litigation; considering that specific avatars will also have financial worth in the Metaverse, these conflicts could increase enormously (*Nitin, 2022*). Therefore, there is a chance that IP's rights will be violated in the Metaverse, concerning works coming from the outside and results coming from within. New original works are being produced within the Metaverse, as witnessed before. Any reproduction or dissemination of this online content by a third party without permission could be considered copyright infringement, as these works are likely to be covered by copyright. The same may occur if a third-party share works created outside the Metaverse—that is, from the real world, such as a painting or photograph—in the Metaverse without the owner of the copyright's consent (or without benefiting from a copyright exception) (*European Innovation Council & SMEs Executive Agency, 2022*).

When we discuss copyrights in the Metaverse context, holders may face several difficulties. First, it might be challenging to police copyright violations on the Metaverse. This problem might be solved by including specialized copyright filters in the Metaverse program. However, it is uncertain whether the Metaverse's creators will be able or willing to include such filters. Second, it will be required to ensure that any licenses that content producers have obtained for copyrightable works before the development of the Metaverse permit usage of the work on this new technology. Users of copyrighted content may require comprehensive rights, which could be provided under a copyright license, given the intricacy of the Metaverse.

Additionally, several plaintiffs asserted that Epic Games, Inc., the company behind the online game "Fortnite," had unlawfully copied their dance techniques and included them in the game (*Levan, 2022*). Besides, there will be copyright difficulties because the Metaverse assets (content) are user-generated. Transferring users' assets from one Metaverse platform to another is a practical issue that needs to be resolved. This is essential because there will be several Metaverse platforms. Such portability and interoperability call for industry standardizations and judicial enforcement (*Cheng et al., 2022*).

Adding to the copyright challenges, intellectual property issues are also quite important. For instance, when a given work in the Metaverse is the outcome of a decentralized collaborative process, it is more challenging to find out who are the producers. Additionally, such ambiguity might alter how courts view fair usage. In the meanwhile, trademark attorneys are concentrating on issues like who should be held accountable when

the identity of the infringement is ambiguous, how trademark dilution can occur in the Metaverse, and whether digital assets should constitute "goods" for purposes of the trademark rules (*Clifford Chance Briefings, 2022*). Numerous studies have been written on whether artificial intelligence objects should be given separate legal personhood, comparing avatars to artificial intelligence systems in the Metaverse. Integrating artificial intelligence tools with Metaverse makes it extremely complicated. It may be practical to give the avatars in the Metaverse the rights and obligations that a human being would have in real world. Doing this task requires assistance from machine learning algorithms which will allow the avatars to complete routine tasks without human assistance (*Cheong, 2022*).

## CONCLUSION

With the help of virtual worlds, new instructional approaches can be developed. This provides the teachers and students with a different way to teach and learn while incorporating both learning models, hybrid and mobile. A new generation of students is now enrolling in universities due to the quick rise in interactive technology use throughout the early years of school. The use of visual imagery and active participation in the learning process has increased recently It also makes it easier for teachers to employ pedagogies like inverted classes and collaborative learning, encouraging flexibility and positive class dynamics. Another thing to keep in mind is the way programming is done. How Metaverses are made, they are scalable, allowing for the incorporation of different ICT resources as well as more interactive elements and the expansion of the created region, which allows for a broader range of applications to be made to suit academic research and leisure needs. Metaverses are scalable, which allows the incorporation of different ICT resources and more interactive elements. They also expand the created worlds, leading to a broader range of applications that meet the academic research needs.

A factor for a novice instructor to consider while interacting with a virtual world is that this may be rapidly altered; all it takes is growing the understanding of ICT tools and their management. As for the student, he adjusts more quickly because he is a digital native and has access to various technological tools that make it easier for him to deal with this resource. In this way, the instructor can make use of this digital resource. As mentioned, with sufficient platform expertise, students and teachers can implement a single study plan, a curricular adaptation, and a methodological adaptation for everyone.

As a result, the design of virtual training spaces must adhere to rigorous and high-quality design standards, just like the design of training spaces in the real world. Doing so, both learners and instructors will maximize their learning in a challenging free environment.

This article also presented the advantages and disadvantage of Metaverse. Briefly advantages are directly related to adding new styles and methodologies of learning that can contribute to better learning environment. On the other hand, the disadvantages are mainly related to infrastructure, ethical concerns and mental health. There are some challenges to incorporating Metaverse in education such as privacy and security issues, device challenges and social and ethical considerations. However, despite of these

challenges, the opportunities offered by Metaverse in education are promising. Every new technology will have some drawbacks that can be later researched and fixed.

## Future research

The Metaverse environment will significantly contribute to guided and unguided learning processes, educational simulations, social learning environments, and remote learning thanks to its engaging, interesting, and amusing characteristics. It is also evident that new chances can be offered to students with impairments and/or those who have difficulties accessing learning resources in terms of providing equal opportunities in education in a Metaverse setting where physical borders have been abolished. However, incorporating any new technology into the classroom could lead to issues with instructional technology. There are issues with, in particular, how to create efficient instructional designs for the Metaverse environment, how learning will occur, and how to conduct an efficient assessment and evaluation. Therefore, it is necessary to develop a new Metaverse pedagogy.

Since Metaverse is a brand-new technological entity, all parties participating in education must accept its technical framework, including students, teachers, administrators, and school boards. Cultural variations, psychological standards, and social conventions will impact how sound Metaverse technology is received.

As a result, significant consideration must be given to using Metaverse in learning processes.

The components and architecture of the Metaverse have not yet been fully developed and implemented because the concept is new. However, the technology that will involve the sense organs in the process needs to be more helpful and less expensive for learners to live and let this experience fully and satisfactorily in learning contexts. This hardware will take some time to mature and become widely used.

National and international cooperation should be created for Metaverse to function appropriately in learning contexts and make good use of the resources allotted for it. To develop the appropriate technology, software, content, and human resources, academic and non-academic parties must collaborate and carefully organize the process.

### Funding
The authors received no funding for this work.

### Competing Interests
The authors declare that they have no competing interests.

### Author Contributions
- Sanaa Kaddoura conceived and designed the experiments, performed the experiments, analyzed the data, performed the computation work, prepared figures and/or tables, authored or reviewed drafts of the article, and approved the final draft.

• Fatima Al Husseiny performed the experiments, analyzed the data, performed the computation work, prepared figures and/or tables, and approved the final draft.

## Data Availability

This article is a literature review and has no raw data.

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
