# Peer review of "The rising trend of Metaverse in education: challenges, opportunities, and ethical considerations"

_PeerJ Computer Science, doi:10.7717/peerj-cs.1252_

## Round 0.1 · original submission · Major Revisions

Please incorporate the reviewers' comments and revise the manuscript.

Reviewer 1 ·

Basic reporting

In the abstract section, it is necessary to briefly include the research method (systematic review-PRISMA).

In the background, the first sentence immediately highlights VR and AR. Before discussing VR and AR, it is necessary to add a brief review of the previous literature on technology in education and its development until the metaverse appears, and then relate it to the inverted pyramid of Maslow's revised edition. This is to clarify the context of the development of education and its relation to the emergence of metaverse. In addition, an explanation of the context of educational development will clarify who the target audience of this article is.

Metaverse in education is an updated and good topic; the systematic review carried out in this manuscript has been well reviewed, but in the discussion section, some are not relevant enough to be discussed. For example, in the “Deepfake and alternate representations of the real world” section (line 633 - 653) and the policies premises section (line 703 - 744). I presume these two discussions are not relevant to educational issues.

The article’s structure including pictures and tables, has met the criteria of PeerJ Computer Science. Figures and tables are relevant to the explanation given in the body text. The English grammar presented already meets the rules of good writing, but there are minor errors (such as commas, brackets, and periods that are misplaced) that need to be corrected (see attachment). Improvements that need to be made also appear in some paragraphs that do not have main and explanatory ideas (For example, lines 1 - 3, 30 - 34, 190 - 201, see attachment for more details). The paragraphs are also not proportional, some are short, but some are too long. It would be better if these paragraphs were presented in a balanced composition and proportion. There are also some mistakes in writing references (see attachment).

Experimental design

In general, this manuscript fulfills the aims and scope of PeerJ Computer Science. Data collection methods have been described in detail and supported by adequate visualization (graphs and tables).
The articles reviewed and their criteria (inclusion and exclusion) are sufficient. The sources of citations need to be added and in some sentences, there are no sources (For example lines 48-50, 90-94, and more can be seen in the attachment).
The coherence between paragraphs is quite good, but the discussion section (lines 277-280) needs to be moved because it is not coherent with the paragraphs before and after it. Linę 431-437 talks about the metaverse and how it relates to devices, so it's best to move to the “Device Challenges” section.

Validity of the findings

The impact of the results of this systematic review has been described in the Future Research section, including the opportunities for replication. Unfortunately, in the conclusion section, not all of the research questions have been answered, especially in Q3, Q4, and Q5.
In the discussion section, it is necessary to focus on the topic of the metaverse and its relation to education only.
Other discussions outside the context of education should be limited and need to be supplemented with literature studies related to the metaverse (eg about the technological knowledge of educators, knowledge of educators, and educational practitioners about the metaverse).

Annotated reviews are not available for download in order to protect the identity of reviewers who chose to remain anonymous.

·

Basic reporting

The report is clear and well-written. There are a few places where some more careful editing could be done (e.g., line 39 has "...Ipads, laptops, and I-pods" -- the authors should standardize on iPad / iPod, or I-pad / I-pod).

The literature references are comprehensive, and context is provided for all references.

The article has a professional structure. With regard to the figures, Figures 5 and 6 are not particularly clear. I was unsure how to interpret the "cone" for both diagrams. Those two figures would probably be my strongest criticism of the paper -- they should be modified to be clearer.

The review is of broad and cross-disciplinary interest, and for those looking for a Metaverse review, this article will suffice (at least for the near future, until things change in unanticipated ways).

The Introduction adequately introduces the subject.

Experimental design

The article seems to be within the scope of the PeerJ Computer Science journal as meta-review / survey.

The research seems rigorous and I have no concerns about the technical or ethical standards with regard to the paper.

The survey is comprehensive and mostly unbiased. The one thing that I was surprised was not covered thoroughly was the commercial ownership of the Metaverse with regard to advertising -- this is one of the biggest criticism's of Mark Zuckerberg's Metaverse, that it will be simply all about advertisements and propaganda, and this was not covered in great detail.

The sources are adequately cited, a nd the review is organized logically.

Validity of the findings

The conclusions are well stated, and do a good job of capturing an overview of the Metaverse in its current state.

This isn't the type of paper that has a well-formed argument, but the goals set out in the Introduction were clearly met.

The authors do point to future work after their conclusion.

Additional comments

This was a pretty good survey, and if I was looking to see the current state of the Metaverse, this is a paper I would use for my research.

---

## Round 0.2 · Minor Revisions

Please incorporate the reviewer's comments.

Reviewer 1 ·

Basic reporting

My previous comment on Research Methods (PRISMA) was followed up and included in the abstract. Abstracts are now more legible and clear. The introduction adequately introduces the metaverse in general and in education. The irrelevant topic, as previously mentioned has been removed and now this manuscript is more readable.

Please check some errors in lines: 153, 154, 294, 295, 399, 746, and 755. Please add some more relevant citations in lines 213 - 222 and 343 - 351; using one source in one paragraph is not appropriate. Check all citations are linked to the bibliography (double-check in lines: 225, 234, 271).

Experimental design

The Survey Methodology is detailed, adequate, and appropriate. All my previous comments are addressed.

Validity of the findings

Findings has focused on discussing the metaverse and its relation to education. Discussion outside the topic of education has been removed. Likewise, the conclusion is in accordance with the formulation of the problem.

Additional comments

Well done.

---

## Round 0.3 · accepted · Accept

This manuscript is ready for publication.

Reviewer 1 ·

Basic reporting

The manuscript has been revised as per my suggestions. Please be considered the paragraph structure, as a paragraph consists of a main idea and supporting ideas; e.gg. line 225 - 237.

Experimental design

The study design is well written and clear.

Validity of the findings

Validity and the findings are well developed and supported by evidences.